# A Computational Evaluation of the Steric and Electronic Contributions in Stereoselective Olefin Polymerization with Pyridylamido-Type Catalysts

**DOI:** 10.3390/molecules28093768

**Published:** 2023-04-27

**Authors:** Olga D’Anania, Claudio De Rosa, Giovanni Talarico

**Affiliations:** 1Scuola Superiore Meridionale, Largo San Marcellino, 80138 Napoli, Italy; 2Dipartimento di Scienze Chimiche, Università degli Studi di Napoli Federico II, 80124 Napoli, Italy

**Keywords:** stereoselective olefin polymerization, DFT calculations, activation strain model, nonmetallocene catalysts

## Abstract

A density functional theory (DFT) study combined with the steric maps of buried volume (%*V*_Bur_) as molecular descriptors and an energy decomposition analysis through the ASM (activation strain model)–NEDA (natural energy decomposition analysis) approach were applied to investigate the origins of stereoselectivity for propene polymerization promoted by pyridylamido-type nonmetallocene systems. The relationships between the fine tuning of the ligand and the propene stereoregularity were rationalized (e.g., the metallacycle size, chemical nature of the bridge, and substituents at the *ortho*-position on the aniline moieties). The DFT calculations and %*V*_Bur_ steric maps reproduced the experimental trend: substituents on the bridge and on the *ortho*-positions of aniline fragments enhance the stereoselectivity. The ASM–NEDA analysis enabled the separation of the steric and electronic effects and revealed how subtle ligand modification may affect the stereoselectivity of the process.

## 1. Introduction

The advent of single-site α-olefin polymerization catalysts has revolutionized the world of polyolefins, as they enable the fine tuning of polymer microstructures in terms of stereoregularity [1], regioregularity [2,3] molecular mass [4,5,6], and polymer properties [7]. Increasing interest in the design of novel nonmetallocene complexes [8] has also opened up a way to further investigate the factors that determine the origin of the stereocontrol of **α**-olefin polymerization promoted by homogeneous systems. Some interesting examples of this are the pyridylamido-Hf catalyst model compounds developed by Coates et al. [9] (**Ia**, **Ib**, [Fig molecules-28-03768-ch001]) and the modification [10] (**IIa**–**IId**, [Fig molecules-28-03768-ch001]) of Dow Chemical systems [11] obtained via high-throughput technologies (**IIe**, [Fig molecules-28-03768-ch001]). Besides propylene, systems **IIc** and **IIe** ([Fig molecules-28-03768-ch001]) have been also employed to polymerize higher α-olefins such as 1-hexene and 1-octene [12], as well as to incorporate functional comonomers [13]. Moreover, block copolymers can be obtained through “chain shuttling” processes, which involve **IIe** in combination with a second catalyst [14,15,16,17]. Density functional theory (DFT) calculations have revealed that **IIa**-**IIe** produce isotactic polypropylenes (iPP), despite the *C*_1_ or *C*_s_ symmetry of the catalyst precursors, through a combination of peculiar aspects that distinguish this class of complexes, which are briefly summarized here. The cationic species deriving from the cocatalyst activation show both the Hf-C_Aryl_ and Hf-C_Alkyl_ bonds into which the first olefin insertion may occur, and it has been demonstrated that the initial insertion takes place at the Hf-C_Aryl_ bond, thus generating the monoinserted active species (Figure 1) [18,19,20,21].

The DFT calculations also found a substantial preference for propene insertion at one of the two diastereotopic sites thus generated (Figure 1) [22], leading to a chain stationary (CS) mechanism (or site epimerization) [23] different from the usual chain migratory (CM) mechanism (the switching of the olefin and the growing chain at each insertion step) [24]. Very recently [25,26], we identified the stereo-electronic factors that induce a CS mechanism by using a combined approach based on DFT calculations, a sterical descriptor (the percentage of occupied volume (%*V*_Bur_) [27,28]), and an activation strain model (ASM) with a natural energy decomposition analysis (NEDA) scheme [29,30], which were applied to polymerization catalysis [31,32].

This family of catalysts deviates from the classical “chiral growing chain orientation” mechanism of stereocontrol [1,33] operating on *ansa*-metallocene [1] and heterogeneous Ziegler–Natta (ZN) systems [33,34], because its stereoselectivity is indeed based on a direct monomer–ligand interaction mechanism [22]. The experimental propene polymerization data (Appendix A) show that the *ortho*-substituents on the aniline moiety (R in [Fig molecules-28-03768-ch001]) play a crucial role in the stereoselectivity, with the catalyst performance also being influenced by the R^1^ and R^2^ substituents on the bridge linking the pyridine and the N-aryl fragment ([Fig molecules-28-03768-ch001]). Although quite far from the active sites, they do affect the enantioselectivity of the complex, pushing the N-aryl ring, and consequently its *ortho*-substituent, closer to the active site thus, enhancing the stereoselectivity of the catalyst through a “buttressing effect” [10]. Furthermore, Hagadorn et al. [35] claimed that a substitution of the C-bridge with a Si-bridge (**IIIa**, **IIIb**, [Fig molecules-28-03768-ch001]) seems to increase the stereoselectivity of propene polymerization. Finally, a replacement of the aryl group (e.g., phenyl or naphthyl) with heteroaryls [36,37] has also been reported. For instance, Pellecchia et al. synthesized a *C*_s_-symmetric Zr(IV) complex which bears a tridentate pyrrolidepyridine ligand (**IVa**, [Fig molecules-28-03768-ch001]) and affords iPP when combined with Al*^i^*Bu_2_H and methylalumoxane [38]. The presence of Al-H alkyl species is necessary for making the complex stereoselective, and the DFT calculations suggest [39] that Al coordinates with the N_pyrrolic_ and that H interacts with the central metal, leading to a sort of “ligand modification”, similar to what happens for pyridylamido Hf catalysts. The original catalyst symmetry is thus altered by the Al*^i^*Bu_2_H coordination, shifting from *C*_s_ to *C*_1_ symmetry. The two diastereotopic active sites, one which is better described by a pyramidal square geometry (site 1) and the other by a trigonal bypiramidal geometry (site 2), select the same propene enantioface, with the unprecedented combination of a “direct ligand–monomer” interaction for one site and a “chiral growing chain orientation” model for the other [39]. 

In this work, we decided to investigate the propene stereoselectivity promoted by the systems of [Fig molecules-28-03768-ch001] to achieve a unified picture, with respect to the (large) spread of the experimental data (Appendix A). We used a combined approach based on DFT calculations, a %*V*_Bur_ analysis, and an ASM–NEDA model to assess the steric and electronic contributions to the propene stereoselectivity for the fine tuning of: (a) the metallacycle size, characterized by a six-membered (**Ia**-**Ib**) and seven-membered (**IIa**-**IIe**, **IIIa**, **IIIb**) ring, respectively; (b) the chemical nature of the R, R^1^, and R^2^ substituents located on the aniline ring and the bridge linking the pyridine and the aniline fragments; (c) the central atom on the bridge by replacing the C with Si atoms (**IIIa**, **IIIb**); and (d) the pyridylamido framework by replacing the aryl with heteroaryl groups (**IVa**). 

## 2. Results and Discussion

The DFT values calculated for the stereoselectivity of the propene polymerization promoted by the systems of [Fig molecules-28-03768-ch001] are summarized in Table 1. They are reported as the differences in the electronic energies (free energies) between the lower 1,2 *si* and 1,2 *re* propene enantioface insertion transition states (TSs), which were calculated in the presence of a solvent contribution (first column, PCM model, see Section 3). Since the ASM–NEDA analysis employs electronic energies in the gas phase (Δ*E*), for the sake of consistency, we also report the differences in the DFT electronic energies (free energies) values in the gas phase (Table 1, second column). Given the findings about the CS mechanism disclosed for the pyridylamido-Hf complexes [22], only the results for the propene insertion at the preferred site are reported for **Ia**-**IIIb**, whereas the energetics for the monomer insertion at both diastereotopic sites are reported for **IVa**. The partitioning of Δ*E*_Tot_ into its contributions obtained through the ASM–NEDA analysis is reported in Table 2 and the details about the decomposition of the Δ*E*_Int_ into all its terms can be found in Appendix A. To simplify the discussion, we also added into Table 2 the ΔΔ*E* between the 1,2 *re* and 1,2 *si* enantiofaces (kcal/mol) obtained by the ASM–NEDA analysis. The effect of the dispersion corrections on the DFT electronic energies is reported in Appendix A, whereas the values of the energetic terms obtained through the ASM–NEDA scheme, without including the dispersion corrections, are illustrated in Appendix A. 

Looking at the ASM–NEDA results in Table 2, we noted that the Δ*E*_Strain_ is, indeed, the main factor for the origin of the stereoselectivity promoted by the analyzed systems. The clear preference for the 1,2 *re* enantioface is only partially compensated by the Δ*E*_Int_ contribution, which instead stabilizes the *si* enantioface (**Ia**-**Ib** being the only exceptions). The further decomposition of the Δ*E*_Strain_ into the two components (Δ*E*_Strain(Mon)_ and Δ*E*_Strain(Cat)_) is highly indicative; systems **II** and **III**, characterized by the formation of seven-membered metallacycles, show the propene deformation between the *si* and *re* insertions (ΔΔ*E*_Strain(Mon)_), which outweighs that of the catalyst (ΔΔ*E*_Strain(Cat)_), thus playing the primary role in the Δ*E*_Strain_ variation, although the two components become similar for **IIe** and **IIIa** (Table 2).

On the contrary, systems **I** and **IV**, which cannot undergo a ligand modification in situ, deviate from this trend, and the ΔΔ*E*_Strain(Cat)_ is the dominant term for the Δ*E*_Strain_ variation (Table 2). This difference may be rationalized by examining the orientation of the growing polymer chain obtained by the DFT calculations, taking system **Ia** as an example. In Figure 1, the optimized geometries for the TSs of the right (Figure 1A) and wrong propene enantioface insertions (Figure 1B) promoted by **Ia** are reported. For such a system, characterized by a six-member metallacycle, there is not enough room to accommodate the bent growing chain [40]; therefore, the catalyst structure distorts and ΔΔ*E*_Strain(Cat.)_ becomes the fundamental contribution to the ΔΔ*E*_Strain_. For the other pyridylamido catalysts (see, e.g., **IIa** TS structures reported in Figure 2), the first C-C bond of the *^i^*Bu group simulating the polymeryl chain is perfectly in *anti* with respect to the methyl group of propene, and is thus bent towards the aryl group. It appears that the metallacycle size does affect the iPP stereoselectivity and the results reported in Table 1 and Table 2 show that **IIa** (Δ*E*(Δ*G*)_Stereo_ = 4.0 (3.1) kcal/mol) is more stereoselective than **Ia** (Δ*E*(Δ*G*)_Stereo_ = 2.4 (1.0) kcal/mol), even if they bear the same R, R^1^, and R^2^ substituents, in agreement with the experimental data [9,10] (Appendix A). As already mentioned, the ASM–NEDA analysis suggests that the ΔΔ*E*_Stereo_ for **Ia** is mainly due to the ΔΔ*E*_Strain(Cat)_ contribution, rather than that of the ΔΔ*E*_Strain(Mon)_ (Table 2)_._ As a matter of fact, the Δ*E*_Strain(Mon)_ term favors the wrong propene enantioface insertion (1,2 *si*) rather than the right one (1,2 *re*) and the greater distortion of the latter may be attributable to the presence of an additional disfavoring interaction between the propene and naphthyl moiety (Figure 1A).

Removing this “penalty” with a larger metallacycle (a seven-member metallacycle shown by **IIa**, Figure 2) forces the naphthyl group to stay further from the olefin and the Δ*E*_Strain(Mon)_ becomes the main factor in the stereoselectivity (Table 2).

The variation in the steric hindrance moving from **Ia** to the **IIa** active sites may be visualized by the steric maps of the corresponding neutral mono-inserted species (Figure 3A,B). The northeast (NE) quadrant is effective for the direct monomer enantioface selection, since it contains the N-aryl ring with its *ortho*-substituents that interact with the “wrong” propene enantioface (1,2 *si*). At the same time, the southeast (SE) quadrant may be responsible for adding the penalty for the “right” propene enantioface (1,2 *re*), whereas the southwest (SW) quadrant contains the metallacycles and the naphtyl group. 

The computed %*V*_Bur_ are consistent with the ASM–NEDA results. In fact, the ΔΔ*E*_Strain(Mon)_ term is the main ΔΔ*E*_Strain_ contribution to the system showing the higher %*V*_Bur_ in the NE quadrant (**IIa**). Furthermore, although the SW quadrants of the maps for **Ia** and IIa have comparable buried volumes (81.2% for Ia and 82% for **IIa**), the SE quadrant for **Ia** has a significantly higher %*V*_Bur_ (38.7%) than **IIa** (33.1%). The greater steric hindrance in the SE quadrant for system **Ia** is due to the closer proximity of the naphtyl ring to the active site, caused by the smaller metallacycle size. Consequently, the aryl fragment occupies the SE quadrant along with the SW quadrant, thus adding a small penalty to the “right” propene enantioface insertion in the case of catalyst **Ia** (Figure 3A). Instead, such a penalty is absent for complex **IIa**, where the naphtyl moiety occupies only the SW quadrant (Figure 3B); therefore, it does not interact sterically with the *re* propene enantioface. 

The %*V*_Bur_ steric maps separated by quadrants allow us also to visualize the effect of the substituents R^1^ and R^2^ on the CCN bridge by comparing **IIa** and **IIc** (Figure 3B,C). Through the DFT calculations, we found that the Δ*E*(Δ*G*)_Stereo_ for system **IIa** is higher than that for **IIc** (Table 1), and the ASM–NEDA analysis reveals that the main reason for this energetic difference is the ΔΔ*E*_Strain_, whereas the ΔΔ*E*_Int_ is quite similar (Table 2). In particular, the ΔΔ*E*_Strain(Mon)_ contribution is larger than the ΔΔ*E*_Strain(Cat)_ term and it increases from **IIc** to **IIa**. Indeed, as already reported by Coates et al., bulky substituents at the C-bridge enhance the stereoselectivity through a Thorpe–Ingold-like “buttressing effect”, as they interact with *ortho*-substituents on the aniline ring, forcing them closer to the olefin [10]. This explains the greater distortion of the monomer for **IIa** rather than **IIc**, but also the longer distance between the propene and *^i^*Pr group on the aniline moiety in the 1,2 *si* insertion TS for **IIc** (Appendix A) with respect to **IIa** (Figure 2). The influence of the “buttressing effect” becomes evident when the north quadrants of the steric maps are examined. In fact, the presence of different substituents at the bridge not only affects the %*V*_Bur_ of the northwest (NW) quadrant, but also the buried volume in the NE quadrant, which contains the aniline moiety. Both the NW and NE quadrants of the **IIc** map (Figure 3C) have a lower buried volume (73.8% and 45.3%) than the analogous of **IIa** (75.7% and 50.5%) (Figure 3B). Furthermore, the substitution of the R substituents with smaller groups (from *^i^*Pr to Me) clearly decreases the stereoselectivity of the propene insertion for both the **Ib** and **IIb** catalysts (Table 1). The corresponding DFT-optimized geometries showing less effective ligand–monomer interactions are reported in Appendix A and the %*V*_Bur_ steric maps are reported in Appendix A.

An interesting modification of the chemical nature of the bridge (replacing the central C atom with a Si one) has been recently published in the literature [35]. Our calculated stereoselectivity for system **IIIa** was not particularly encouraging with respect to the analogous system **IId** being the energetic (free energies) difference between the two TSs, with facial selectivities of 2.5 (2.2) kcal/mol and 3.8 (3.1) kcal/mol, respectively. The smaller CCN angle in complex **IId** (Figure 4A,B) forced the *^i^*Pr groups on the N-aryl ring closer to the active site, increasing the steric ligand–monomer interactions (3.61 Å) with respect to the Si-bridged system (Figure 4C,D) (3.88 Å). The preferred configuration of the growing chain was similar for **IId** and **IIIa**, and in both cases, we noted the absence of the chiral configuration of the growing chain [41]. The ASM–NEDA analysis (Table 2) clarified these findings, especially the ΔΔ*E*_Strain(Mon)_, which represents the principal contribution to the ΔΔ*E*_Strain_ and is higher for system **IId** (4.7 kcal/mol) compared to system **IIIa** (3.5 kcal/mol). However, it has been claimed that the suitable modification of silicon-bridged pyridylamido-hafnium complexes may effectively reach a higher stereoselectivity, and when we performed DFT calculations on the cyclo-(CH_2_)_4_Si bridged complex (**IIIa**) employed by Hagadorn [35], we found that **IIIb** was more stereoselective than **IIIa** (Table 1). The presence of a cyclic substituent on the Si-bridge rather than two methyl groups must be regarded as the reason why **IIIb** produces a more stereoregular iPP than **IIIa**. As a matter of fact, the –(CH_2_)_4_- substituent on the bridge makes the complex more constrained, reducing the C-Si-N angle, thereby forcing the N-aryl fragment closer to the monomer (Figure 5). 

In agreement with such statements, the %*V*_Bur_ steric map analysis revealed that there were no notable differences between systems **IId** and **IIIa** (Figure 3D,E), whereas **IIIb** (Figure 3F) showed larger %*V*_Bur_ values in the NW and NE quadrants, in agreement with a higher stereoselectivity. The ASM–NEDA results (Table 2) confirmed that the enhanced stereoselectivity of this complex was due to the larger variation in the Δ*E*_Strain(Mon)_ between the 1,2 *si* and 1,2 *re* insertions, and a quick visualization of the monomer deformation is reported in Appendix A. 

Interestingly, an even higher isotacticity is predicted for the polymerization promoted by the Dow system **IIe** (Table 1); the unsymmetrical substitution of R^1^ and R^2^ on the C bridge increases the ΔΔ*E*_Strain(Cat)_ contribution (with respect to **IIa**), although with a slight decrease in the ΔΔ*E*_Strain(Mon)_ (Table 2). The DFT-optimized geometries for the **IIe** TSs are reported in Appendix A and the %*V*_Bur_ steric map is reported in Appendix A. 

As a final step, we focused on the role of the C_Aryl_ bond (Figure 1) in the pyridylamido framework, replacing the aryl with heteroaryl groups (**IVa**). The presence of a heteroatom in place of the C_Aryl_ bond avoids the ligand modification in situ and the experimental propene isotacticity promoted by **IVa** can be explained by the formation of diastereotopic active sites following the Al*^i^*Bu_2_H coordination (Appendix A). Interestingly, site 1 and site 2 select the same propene enantioface and the stereoselectivity is dictated by the “direct ligand–monomer” for site 1 (Appendix A) and the “chiral growing chain orientation” model for site 2 (Appendix A) [39]. The ASM–NEDA results for **IVa**, although unraveling the subtle differences for the Δ*E*_Int_ contribution at each site, are substantially similar to the ones discussed for **Ia** (Table 2). This demonstrates the relevance of the ΔΔ*E*_Strain(Cat)_ as the main term for the Δ*E*_Strain_ variation and as the origin of isotactic propagation. The loss of the additional contribution of the ΔΔ*E*_Strain(Mon),_ is, in the end, the main reason for the lower stereoselectivity calculated for **IVa** with respect to **IIa**, **IIe**, **IIIa**, and **IIIb** (Table 1), and the way to increase this stereoselectivity is an unsymmetrical substitution with bulky substituents on the bridging methylene atom [37] analogously to system **IIe**.

## 3. Methodology

The DFT calculations were performed by using Gaussian16 programs [42]. B3LYP hybrid functional [43,44] was employed in conjunction with a polarized split-valence basis set (SVP) for H, C, N, Si, O [45] and LANL2DZ basis and pseudopotential for the metal [46]. The stationary points were characterized using vibrational analyses and these analyses were also used to calculate zero-point energies and thermal (enthalpy and entropy) corrections (298.15 K, 1 bar). Improved electronic energies were obtained from single-point calculations using the TZVP basis set for H, C, N, Si, O [47] and the SDD basis set and pseudopotentials [48] for Hf and Zr (an f function with an exponent of 0.5 was added). The dispersion corrections (EmpiricalDispersion=D3 in the Gaussian package) [49] and solvation contribution (PCM model [50], toluene) were evaluated, thus obtaining Δ*G* (B3/D3/TZVP/PCM) values. With the ASM model proposed by Bickelhaupt [29,30], we partitioned the Δ*E*_Tot_ into the Δ*E*_Strain_ and Δ*E*_Int_ components, where the former was the energy associated with the reactant deformation required to achieve the geometries necessary for a reaction, and the latter was the energy associated with the strength of their reciprocal interactions. Furthermore, the Δ*E*_Strain_ contribution was additionally expressed as the sum of the strain of the cationic active species bearing the growing polymer chain (simulated by an *^i^*Bu group), Δ*E*_Strain(Cat)_, and the propene monomer, Δ*E*_Strain(Mon),_ calculated with respect to the optimized geometries of each species. By using the NEDA [51] approach, we also decomposed the Δ*E*_Int_ into its terms, which are: the classical electrostatic interaction (ES), polarization interaction (POL), charge transfer (CT), exchange correlation interaction (XC), and deformation (DEF), which represents the energy required to deform the wavefunction of a fragment in the presence of all the other fragments. The Δ*E*_Tot_ partitioning into its contributions through the ASM–NEDA model is reported in Figure 2.

The NBO version 7 software, coupled with Gaussian16 (in conjunction with the TZVP basis set for H, C, N, O, and Si and the SDD basis and pseudopotentials for Hf), was used to perform the NEDA calculations. The steric maps and the percentage of buried volume for each quadrant (%*V*_Bur_) were computed employing a modified version of the SambVca package [27,28] and the whole computational approach has already been tested on the olefin polymerization catalysis [52]. The steric maps for the neutral mono-inserted species (**IIa**-**IIe** and **IIIa**-**IIIb**) or the neutral catalytic precursors (**Ia**, **Ib** and **IVa**) were created by setting the transition metal as the center of the sphere, whose radius was set to 3.5 Å.

## 4. Conclusions

In conclusion, in this work, we analyzed the origin of the stereoselectivity for propene polymerization promoted by pyridylamido-type catalysts. This catalyst class have emerged within the nonmetallocene family because it allows the formation of block-copolymers [53,54,55,56,57] via living and chain shuttling processes [14], and such molecular architectures are not accessible to *ansa*-metallocenes [58] and heterogeneous ZN systems [59,60]. The mechanism of stereocontrol in olefin polymerization has been proven to be different from the one proposed for *ansa*-metallocenes [1,33], as well as for octahedral nonmetallocene ligands [33,61]. We used a combined DFT/%*V*_Bur_/ASM–NEDA approach for the computational assessment of the steric and electronic contributions to rationalize the ligand structure/polymer microstructure. By using this methodology, we clarified the effect of the metallacycle size, the chemical nature of the bridge linking the pyridine moiety to the N-aryl fragment, the R, R^1^, and R^2^ substituent effects, and, finally, the role of aryl and/or heteroaryl groups. It is worth stressing that the DFT calculations were consistent with the experimental data: hindered substituents at the bridge and the *ortho*-positions on the aniline fragments may increase the stereoselectivity of propene polymerization, as well as moving from a carbon- to silicon-bridge. Analogously, the %*V*_Bur_ steric maps enabled an easy visualization of the ligand steric hindrance, still reproducing the experimental trend. However, a better understanding of the interplay of these steric and electronic contributions was achieved by the ASM–NEDA analysis and its Δ*E*_(Tot)_ energy decomposition into the Δ*E*_Strain_ and Δ*E*_Int_ contributions. The former term is more relevant than the latter for the origin of stereocontrol, and the additional partitioning of the Δ*E*_Strain_ into the Δ*E*_Strain(Cat)_ and Δ*E*_Strain(Mon)_ revealed how subtle ligand modification (see, e.g., the metallacycle size and the asymmetric R^1^ and R^2^ substitution on X atom, Figure 1) increases the propene stereoselectivity. Overall, the complete DFT/%*V*_Bur_/ASM–NEDA approach, is, in our opinion, a powerful tool for the fine tuning of catalyst design/polymer properties [62].

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
