# Peer review of "A Computational Evaluation of the Steric and Electronic Contributions in Stereoselective Olefin Polymerization with Pyridylamido-Type Catalysts"

_molecules, 2023, doi:10.3390/molecules28093768_

Round 1

Reviewer 1 Report

In this study, a series of non-metallocene catalysts for olefin polymerization are investigated by density functional theory calculations. The computational results are well analyzed by buried volume and energy decomposition approaches, aiming to clarify the origin of the stereoselectivity of olefin polymerizations. As a whole, the manuscript is well-written, the results are well-analyzed, and the conclusions are reasonably supported by the computational results. I recommend publication of this work.

Minor issues:

1. Page 3 line 85 and line 106, wrong citation of ref. 22 and 23.

2. Page 4 line 127, electronic “energies”. The word “energies” is missing.

Author Response

Our reply:

We thank the reviewer for the nice comments. The minor issues have been updated.

Reviewer 2 Report

The manuscript ‘Computational Assessment of Steric and Electronic Contributions in Stereoselective Olefin Polymerization by Nonmetallocene Catalysts’ is the trial to analyze the influence of structural elements  for the stereoselectivity of propene polymerization promoted by nonmetallocene systems using computational approach. The paper sounds interesting for the potential reader, but some impeovements are needed before publications in ‘Molcules’ journal.

COMMENTS:

The abstract should be rewritten. There is a need to add the motivation of the research, the goal and 2-3 sentences regarding the main findings. The description of the methodologies should be moved to introduction section.

The English language should be checked from grammar point of view

Ln 34 – add reference to Chart 1

LN 38 – see comment for Ln 34

Ln 82 - … deeply ‘analyzed’ …?

Ln 94 – Computational details section should be replaced by the word ‘Methodlogy’ and should be visualized as a pipeline to make the description understandable by the potential reader.

The introduction – clearly define the scientific impact of the research. The description regarding different methods for he propene stereoselectivity should be added as well as assessment issues.

The description of all materials, software, hardware, statistical tools should be added as well as clear description of the analyzed experimental data set (link to the date will be preferred as well)

The statistics should be added for obtained results, as well as a computational aspects.

Figure 1 – 2 – data for systems IIe should be added

Figure 3 – the description of the scale is needed

In the discussion section the comparison between method used for the assessment is needed. Additionally the importance of the obtained results should be added. The advantages and disadvantages of the analyzed systems regarding introduced assessment should be described and analyzed as well. Results should be described in more understandable and systematic way.

Steric maps (fig and 6) should be presented for all analyzed systems (one figure) as well as DFT optimized geometries. This can make the analysis of the results easier.

The conclusion should be rewritten. Please include main findings corresponding to the title – computational assessment. The conclusion should consist of clear description of the added value of the research. Current version would confuse the reader.

Reviewer 3 Report

The work of D’Anania, De Rosa and Talarico reports a computational study which goal is the in-depth understanding of the reasons for the different stereoselectivity observed experimentally in olefin polymerization reactions catalyzed by several pyridylamido-type complexes of Hf and Zr. The analysis is based on DFT calculations combined with the activation strain model (ASM) and analysis of the percentage of buried volume (%Vbur). This work is a continuation of a previous article reported by two of the current authors (reference 22). Several compounds (Ia, Ib, IIa, IIb and IIc in Chart 1) were analyzed in the previous article and the new additions of the current work, besides expanding the target to more complexes (IId, IIe, IIIa, IIIb and IVa), is the utilization of the ASM and determination of the %Vbur, not performed in the previous one for any of the compounds. This new study allow the authors to evaluate separately the role played by steric and electronic factors in the stereoselectivity observed in olefin polymerization as a function of the catalyst employed.

The manuscript is well written, the introduction and references show a balanced assessment of the current literature and the conclusions are supported by the computational studies reported. The reaction studied, olefin polymerization, is an important process and the knowledge of the mechanisms and factors affecting them can have a big impact in the design of better catalysts for this transformation. However, in my opinion, the article presents low originality and novelty since the main conclusions of this paper were previously know, it is too specialized making it not interesting for a broad audience since the study reported does not contribute new perspectives to the field. This can be realized reading the conclusions section where no new insights are offered. Therefore, I do not recommend that this paper is accepted for the publication in Molecules and find it more suitable for a more specialized Journal.

In any case, computational details have been adequately described, the work seems to have been done carefully and is reported in detail. The results appear to have quality and provide some valuable parameters that offer important aspects for the understanding of the reactions of olefin polymerization catalyzed by pyridylamido-type complexes of Hf.

Several minor points are summarized below:

-          Along the entire main text, it is said that ASM-NEDA analysis is performed, however, the data from the NEDA analysis is not discussed in any part of the text and only the results from the ASM study are reported and discussed. The NEDA results are in the Supplementary Materials (SM) but not in the main text. Thus, I would recommend to let the NEDA data in the SM and remove it from the text saying that ASM analysis is performed (instead of ASM-NEDA). Also Scheme 2 is not really needed if the results from the NEDA model are not discussed.

-          Line 138: In table 1, values correspond to differences in electronic (or Gibbs energies). Thus, I recommend the following: “... we report also the differences in DFT electronic energies…“

-          Values in table 1 are slightly different from that previously reported in reference 22, especially for compound IIa. A sentence indicating the reason would be desirable, probably better in the SM.

-          Table 1 show differences in the electronic (or Gibbs) energies between the TS for the 1,2-si and 1,2-re insertions. In addition, it would be interesting to have the values for the activation energies and activation Gibbs energies for any of the insertions (1,2-si or 1,2-re; the other can be derived with the differences shown). This can help the reader to see the differences in the activation parameters using the different catalysts.

-          Table 1 show two set of values for the complex IVa since it is reported for both diasterotopic sites. However, it is not said in the main text with one is kinetically preferred nor which is the structure of any of them. The structures are reported in the SM and, therefore, I would recommend to give the two possibilities there and keep only the values for the preferred one in Table 1. In addition, the discussion of the data corresponding to this complex, and also for IIe, (Lines 304-316)  are too reduced and contrast to the rest of complexes for which an extensive analysis is performed in the article.

-          DFT calculations have been performed using empirical dispersion corrections (D3) and an analysis of the effects of the inclusion of them would be desirable. Selected calculations without them would give information related to the attractive interactions between the through space interactions between different fragments in the molecules.

Round 2

Reviewer 2 Report

Authors addressed most of the coments but there are still one important issue that has to be clarified. The title is ‘Computational Assessment of Steric and Electronic Contributions in Stereoselective Olefin Polymerization by Nonmetallocene Catalysts’, the goal is ‘deeply investigation of  the propene stereoselectivity promoted by the describe systems’, but the abstract and conclusion section focused mainly on computational side of the analysis almost completely omitting the assessment section. Lacj of consistency can confude the reader. Having in mind above, to have the paper ready for publication, authors have to decide: change the title, as e.g. evaluation of new computational assessment methodology of Steric ….’ or improved abstract and conclusions section by main findings regarding obtained results of Steric and Electronic Contributions in Stereoselective Olefin Polymerization by Nonmetallocene Catalysts .

Author Response

We accepted the Reviewer 2 concerns. We changed the title as suggested and we have also improuved both the Abstract and the Conclusion.